# Identification of Eco-Climatic Factors Driving Yields and Genotype by Environment Interactions for Yield in Early Maturity Soybean Using Crop Simulation

Chloé Elmerich [1,2,*] , Guénolé Boulch [1], Michel-Pierre Faucon [1] , Lyes Lakhal [3] and Bastien Lange [1,*]

1    AGHYLE, UniLaSalle, 19 rue Pierre Waguet, 60000 Beauvais, France
2    Lidea Seeds, 6 chemin de Panedautes, 31700 Mondonville, France
3    Transformations & Agro-Ressources Research Unit–ULR 7519, UniLaSalle, 19 rue Pierre Waguet, 60000 Beauvais, France
*    Correspondence: chloe.elmerich@unilasalle.fr (C.E.); bastien.lange@unilasalle.fr (B.L.)

**Abstract:** Deploying crops in regions bordering their initial distribution area requires adapting existing cultivars to particular environmental constraints. In this study, we revealed the main Eco-climatic Factors (EFs)—climatic factors recorded over specific phenological periods—impacting both yields and Genotype by Environment Interactions (GEI) for yield in early maturity soybeans (*Glycine max* (L.) Merrill) under high latitudes. A multi-year (2017–2021) and multi-environment (*n* = 112) database was built based on the official post-inscription French soybean trial network "SOJA Terres Inovia-GEVES-Partenaires". Yields of 57 cultivars covering MG00 and MG000 maturity groups were considered. For each environment, 126 EFs were calculated using a Crop Growth Model (CGM) based on observed weather data and simulated developmental stages. Partial Least Square (PLS) regression analyses using the Variable Importance in Projection (VIP) score were used to sort out the most relevant EFs for their impact on yield levels on the one side and on GEI for yield on the other side. Our results confirmed that yield levels for both maturity groups were greatly influenced by climatic factors from the seed filling phenophases, mainly End of Pod to Physiological Maturity. The cumulative potential evapotranspiration during the End of Pod to Physiological Maturity period was the main EF affecting yield levels positively for both maturity groups (VIP = 2.86; $R^2$ = 0.64). Interestingly, EFs explaining yield levels strongly differed from those explaining GEI, in terms of both climatic factors and phenophases. GEI were mostly influenced by climatic factors from First Flower to End of Pod; these factors were maximum temperatures and solar radiation intensity. Cold stress from Sowing to First Seed also appeared to be a critical driver for GEI in MG00 soybeans. The contrasted responses of several cultivars to the main GEI-drivers highlighted a potential genetic variability that could be exploited in early maturity soybean breeding. This study revealed the complexity of GEI ecophysiology, and our results should help breeding strategies to deliver germplasm that outperforms the existing genetic material for expanding the crop to northern European regions.

**Keywords:** CROPGRO; crop growth model; DSSAT; genotype × environment interactions; phenophase; protein crop; soybean breeding

## 1. Introduction

The continuous improvement of crop adaptation to the environment is essential for maintaining crop productivity in the context of increasing food demand [1]. Furthermore, crop diversification is essential to more sustainable agriculture [2]. This diversification could be implemented either by the adaptation of species that are not cultivated yet, requiring considerable breeding efforts, or by the deployment of crops that are at the limit of their distribution area [3,4]. The first challenge of expanding crops into a new cultivated area is to assess the yield potential and stability of the existing germplasm under new environmental conditions [5].

Concerning yield potential, the most influential factors regularly cited are $CO_2$ concentration, solar radiation and temperatures [6]. However, the impacts of those factors are frequently studied individually and are rarely ranked and prioritised [7–10]. Furthermore, the assessment of climatic factors influencing relative cultivar performances across environments, i.e., Genotype by Environment Interactions (GEI; Table 1 includes a description of abbreviations and acronyms used in this paper), constitutes an important lever to develop relevant breeding strategies [11].

**Table 1.** List of abbreviation and acronyms used in the paper.

| Abbreviation | Definition | Abbreviation | Definition |
|---|---|---|---|
| CGM | Crop Growth Model | MET | Multi-Environment Trial |
| cv. | Cultivar name | MG | Maturity Group |
| DSSAT | Decision Support System for Agrotechnology Transfer | PLS | Partial Least Square regression |
| EF | Eco-climatic Factor | PY-MG00 | PLS meta-analysis using pairs of years for MG00 |
| EMFI | Emergence to Flower Induction | PY-MG000 | PLS meta-analysis using pairs of years for MG000 |
| EPPM | End of Pod to Physiological Maturity | RMSE | Root Mean Square Error |
| FFFP | First Flower to First Pod | SEM | Sowing to Emergence |
| FIFF | Flower Induction to First Flower | SY-MG00 | PLS meta-analysis using single years for MG00 |
| FPFS | First Pod to First Seed | SY-MG000 | PLS meta-analysis using single years for MG000 |
| FSEP | First Seed to End of Pod | VIP | Variable Importance in Projection |
| GEI | Genotype by Environment Interactions | | |

Understanding GEI is key to maximise genetic gain in plant breeding [12–14]. Multi-Environment Trials (MET), from breeding programmes or official trials, constitute a source of information to assess GEI [15]. However, the environmental context is often too poorly characterised in MET datasets to link crop growth conditions to ecophysiological processes. Crop Growth Models (CGM) as a function of time, environment (climate, soil and management) and genetics could be used to describe plant growth and development [16–19]. The environmental context can then be described by combining developmental stages with duration, temperature, solar radiation and stress factors such as heat, cold, drought, anoxia and nitrogen, generating Eco-climatic Factors (EFs). Therefore, understanding GEI could be tackled by revealing the main EFs driving GEI [16,20]. In this paper, the main EFs driving GEI will be referred to as GEI-drivers. Specific statistical methods are needed to reveal them; some have already been developed in the literature, and among them, Partial Least Square regression (PLS) seems the best suited [21,22].

In the context of the European protein deficit, leguminous crop deployment has been accelerated [23,24]. Among legumes, soybean (*Glycine max* L. (Merrill)) might be a good candidate as a major oil-protein crop for both food and feed [25]. Moreover, introducing soybean could be particularly effective for diversifying cropping systems, helping to break down disease, pest and weed cycles and limiting the use of nitrogen fertilisers [26]. Soybean is grown mainly in Brazil (tropical regions) and in the United States (temperate regions) because of the optimal growth conditions and historical breeding efforts. In western Europe, soybean is essentially grown in northern Italy and southwestern France [27]. To develop soybean in European regions, northern expansion is preferable, considering the negative impacts of climate change on the Mediterranean Basin as maximum temperatures increase and summer water shortages forbids irrigation [28].

Because soybean is a thermophilic short-day plant sensitive to photoperiod, the expansion of this crop to northern areas first requires the adaptation to long days [29]. This expansion has resulted in the release of early and very early cultivars belonging to Maturity Groups MG00 and MG000, respectively. Soybean physiology and, by extension, yield potential are known to be impacted by various abiotic factors [30]. Low temperatures (under 15 °C) are considered as a chilling stress in soybean and have been reported to affect plant growth and pod setting [31–34]. Water deficits and drought stress have also been largely identified as major yield-limiting factors [35–37]. These factors have different effects

on crop performance, depending on the growth type: indeterminate or determinate, as vegetative and reproductive phases sometimes overlap [38,39]. The reproductive phase, particularly from pod setting to physiological maturity, has been well documented and perceived to be the most critical one [30,40–42].

Recent studies have addressed different aspects of soybean northern expansion in Europe: predicting soybean phenology [43], simulating emergence and germination [44], identifying agro-economic potential [45] and determining major environmental covariates influencing simulated and observed yield levels [46,47]. However, concerning the understanding of GEI, only the genetic dimension has been investigated by identifying genetic regions linked to GEI for soybean grown in the US Corn Belt [48]. No focus has yet been put on the environmental dimension of GEI. A sufficient early soybean MET database is needed to unravel the GEI variance resulting from the contrasted contributions of a large number of EFs [20]. Therefore, the objectives of this study were to (i) reveal the main EFs driving GEI for yield in early maturity soybean in northern European regions, (ii) compare the GEI-drivers to the main EFs impacting yields and (iii) illustrate genotypic responses to the GEI-drivers. Our results are expected to provide enlightenment for breeders to accelerate soybean genetic gain under high latitudes in northern Europe.

## 2. Materials and Methods

### 2.1. Experimental Dataset

#### 2.1.1. Multi-Environment Trials Data Source

The French technical institute for oil and protein crops, called Terres Inovia, has produced each year since 2017 a public synthesis of statistically validated post-registration soybean cultivar trials from the network *SOJA Terres Inovia-GEVES-Partenaires* used for varietal recommendation (https://www.myvar.fr/resultats/campagne-177.html). Two Maturity Groups (MGs) were investigated in our study: very early maturity MG000 and early maturity MG00. The trials were located in France between the forty-fifth and the fiftieth north parallels. France's climate type is considered temperate, with both oceanic and continental influences [49]. The trials were distributed mostly in the north, the centre and the east of France, following a temperature gradient and conducted independently by MG (Figure 1). The synthesis provided by Terres Inovia for each trial included the trial location, the cultivars yield and the cultural practices (sowing dates and irrigation amount).

The database regrouped 112 environments, a combination of the location, year and maturity group. The environments are distributed over five years (2017 to 2021). A total of 57 cultivars were tested (29 from MG00 and 28 from MG00). The detailed distribution of cultivars over the five years can be found in Supplementary Table S1.

#### 2.1.2. Weather and Soil Data Source

Gridded (25 km/25 km) meteorological data were extracted from the Agri4Cast Resources Portal, available at the European Joint Research Centre (JRC) website. The closest gridded point (in kilometres) was attributed to each trial location. Daily minimum and maximum temperatures (°C), daily precipitation (mm) and solar radiations (MJ m$^{-2}$ day$^{-1}$) from 1 January to 31 December were extracted. Soil parameters extracted from raster files produced by the European Soil Data Centre (ESDAC) [50] were used to characterise each soil location (1 km/1 km grid) using QGIS3 (v. 3.14.1). Clay content (%), silt content (%), sand content (%), gravel (%), organic carbon content (%), total nitrogen content (%) and bulk density (g cm$^{-3}$) from topsoil and subsoil were extracted as well as the depth available to roots (cm).

### 2.2. DSSAT Simulations and Eco-Climatic Factors Calculation

Eco-climatic Factors (EFs) consisted of climatic variables calculated between two developmental stages (i.e., phenophases) over the entire crop cycle. The CROPGRO-soybean model—DSSAT v4.7.5 (Decision Support System for Agrotechnology Transfer) [51–53] was

used to calculate the phenophases needed for EFs calculation. The CROPGRO model demands weather data, soil parameters, crop management and genotypic inputs [52].

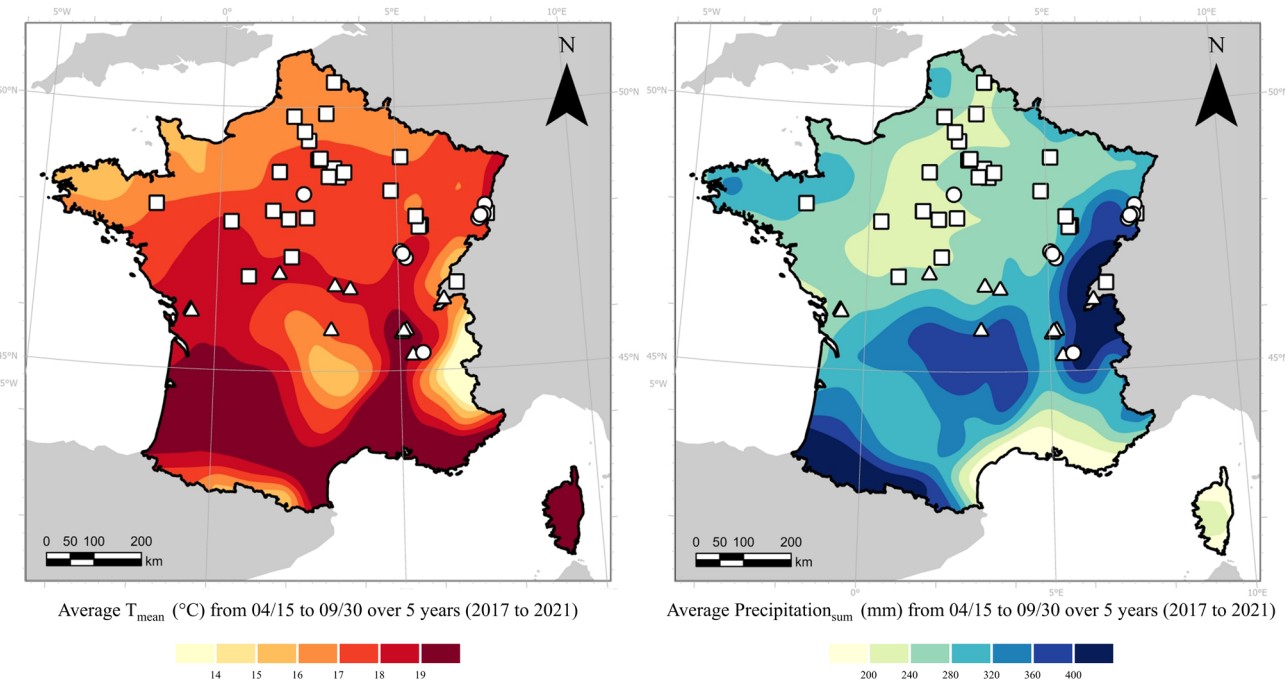

Average $T_{mean}$ (°C) from 04/15 to 09/30 over 5 years (2017 to 2021)    Average Precipitation$_{sum}$ (mm) from 04/15 to 09/30 over 5 years (2017 to 2021)

**Figure 1.** Geographical repartition of the Multi-Environmental Trial (MET) under French climatic conditions. The MET is constituted of 112 environments (a result of the unbalanced combination of 60 locations and five years). Both maps indicate the trials' positioning over the five years (2017 to 2021) by tested maturity: MG000 (□), MG00 (Δ) and both (○). The maps show a colour gradient of mean temperature (left) and cumulative precipitation (right) over the five years, calculated on the average growing period from 15 April to 30 September (data source: Joint Research Centre meteorological data).

### 2.2.1. CROPGRO-Soybean Model Inputs

Daily weather data and soil parameters were used to characterise each environment (see Section 2.1.2). A minimum set of crop management inputs was used to run the simulations: the sowing date and the irrigation scenario. Recorded sowing dates ranged from 04/17 to 06/01. Recorded irrigation amount ranged from 0 mm to 330 mm. Sowing parameters were set at 65 plant m$^{-2}$ with 35 cm spacing and 3 cm depth which are the usual farmers' practices in the area recommended by the technical institute, Terres Inovia (https://www.terresinovia.fr/p/guide-soja).

Concerning the genotypic inputs, ideally, the growth stages of each genotype in each environment should be measured or simulated. Indeed, there is some phenotypic variability between cultivars' phenophases [43,54]. However, phenophases had not been measured and the simulation required the calibration of each cultivar which was not feasible [19]. Moreover, the purpose was to use environmental explanatory variables (referred to as EFs in our study) to explain GEI for yield. These variables have to be unique for an environment and the same for all the tested cultivars in this environment. This approach was developed by Vargas et al. (1998) [22] on wheat crop. The CROPGRO-soybean model uses a set of genetic coefficients to describe a cultivar. The genetic coefficients for phenology prediction are highly related to cultivar photoperiod sensitivity [55]. The differences in phenophases are therefore less between cultivars belonging to the same maturity group [54]. The model has been tested in northern France by Boulch et al. (2021) [46] and showed very good phenological predictions using generic MG000 and MG00 cultivars available in the model. Thus, in the 67 environments where MG000 were grown, the MG000 generic

genetic coefficients were used whereas in the 45 environments where MG00 were grown, the MG00 generic genetic coefficients were used (Supplementary Table S2).

A total of 112 simulations were performed corresponding to the 112 environments of the MET, i.e., the location, year and maturity group combination.

### 2.2.2. CROPGRO Outputs and Eco-Climatic Factors Calculation

Using CROPGRO-soybean model, soybean developmental stages were simulated. These stages were used to divide the crop cycle into seven phenophases, i.e., periods between two consecutives stages: Sowing to Emergence (SEM), Emergence to Flower Induction (EMFI), Flower Induction to First Flower (FIFF), First Flower to First Pod (FFFP), First Pod to First Seed (FPFS), First Seed to End of Pod (FSEP) and End of Pod to Physiological Maturity (EPPM). In the CROPGRO-soybean model the phenology simulation is based on daily parameters: preceding stage, photoperiod and temperature functions, combined with photoperiod, temperature, water and nitrogen sensitivities [52,56].

Five major categories of environmental variables were calculated for each phenophase: period duration (number of days), temperature (average minimum temperatures in °C, number of days below 10 °C, number of days below 15 °C [31–34], average maximum temperatures in °C, number of days above 30 °C, number of days above 34 °C, average mean temperature in °C and thermal amplitude in °C, defined as the difference between the maximum and minimum temperature), water (cumulative precipitation in mm, evapotranspiration and potential evapotranspiration in mm using Priestley-Taylor/Ritchie method), solar radiation (photoperiod in hours, cumulative daily solar radiation and average of solar radiation in MJ m$^{-2}$, photothermal quotient defined as the ratio of solar radiation on heat units) and stresses (water and nitrogen stress indices), calculated by the DSSAT model and based on the supply demand ratio [57]. Finally, each trial was described by 126 eco-climatic factors that were used to explain both yields and genotype by environment interactions.

### 2.3. Partial Least Square Regression Analysis for the Selection of Eco-Climatic Factors

Partial Least Squares regression (PLS) is a bilinear regression method in which data are decomposed using latent variables: response variables $y_i$ are a linear combination of explanatory variables $x_i$. This model is well adapted for analysis, having the number of explanatory variables (*p*) exceeding the number of observations (*n*) and explanatory variables that are mutually correlated [58]. In PLS, the *p* explanatory variables are stored in the matrix $X = (x_1, \ldots, x_p)$, and the *v* response variables are stored in the matrix $Y = (y_1, \ldots, y_v)$, where *v* is the number of response variables that can be unique or multiple. Each $x_1, \ldots, x_p$ and $y_1, \ldots, y_v$ vectors must have *n* dimensions. An $x_i$ vector having a variance equal to 0 is excluded.

PLS analysis generates statistical parameters that can be used to sort out the most important explanatory variables [59]. Among them, the Variable Importance in Projection (VIP) can serve as a selection criterion. A VIP was calculated for each EF. The average of all squared VIPs is equal to 1, so 1 is commonly used as a threshold for selecting a set of relevant variables [60]. This criterion is very reasonable for discarding irrelevant variables, but in our case, this threshold was increased to keep the most impacting variables (see Sections 2.3.1 and 2.3.2). All analyses were performed under R software (version 4.0.5).

### 2.3.1. Yield Analysis

For the yield analysis, the X matrix contained the 126 calculated EFs for each trial and the Y matrix contained the yields (*v* = 1) calculated as the average of all cultivar yields grown at each trial. To avoid variable dimension issues, the X matrix was centred and scaled. PLS analyses were performed on MG00 trials (45), MG000 trials (67) and trials combining both maturity groups (112). The number of components selected in each PLS was defined based on minimising the Root Mean Square Error (RMSE) using cross-validation [61]. A VIP of 1.9 was used as the threshold for retaining the most relevant yield contributing EFs.

2.3.2. Genotype by Environment Interactions Analysis

In the GEI analysis, the X matrix contained the same EF calculated for yield analysis, and the Y matrix contained the yield per cultivar per trial ($v > 1$). The X and Y matrices were centred and scaled. Because the trials were unbalanced (i.e., not all the cultivars were tested every year) and the MG00 and MG000 trials were independent, separate PLS were conducted. A total of 25 PLSs were run, combining maturity groups (MG00, MG000) with years (2017 to 2021) and maturity groups with pairs of years (2017 and 2018, 2017 and 2019, 2017 and 2020, 2017 and 2021, 2018 and 2019, 2018 and 2020, 2018 and 2021, 2019 and 2020, 2019 and 2021 and 2020 and 2021). For each PLS, the number of components was selected based on the evolution of $R^2$ between the observed and the calculated Y matrix, using the highest increase between two $R^2$ to define the number of components to retain.

PLS results were analysed by maturity group and single years or pairs of years, leading to four meta-analyses: single years for MG00 (SY-MG00), single years for MG000 (SY-MG000), pairs of years for MG00 (PY-MG00) and pairs of years for MG000 (PY-MG000). A score was defined per EF per meta-analysis to summarise the five (single-year) or ten (pair of years) VIP values using the following equation (1), where $\text{Score}_{\text{EF i,n}}$ is the score calculated for the $i$th EF with $i \in [1;126]$ in the $n$th meta-analysis $n \in [1;4]$; $\text{VIP}_{\text{EF } i,j}$ is the VIP obtained for the $i$th EF in the $j$th PLS with $j \in [1;5]$ for single-year meta-analyses; or $j \in [1;10]$ for pair of years meta-analyses. $\text{Scores}_{\text{EF i,n}}$ of 1.18, 1.17, 1.21 and 1.21 were used as a threshold, representing 10% of the EF having the main influence on GEI.

$$\text{Score}_{\text{EF } i,n} = \sum_{j=1}^{4 \text{ or } 6} \frac{\text{VIP}_{\text{EF } i,j}}{\sum_{i=1}^{126} \frac{\text{VIP}_{\text{EF } i,j}}{126}} \tag{1}$$

## 3. Results

### 3.1. Main Eco-Climatic Factors Impacting Yields

As the Eco-climatic Factors (EFs) detected by the PLS analyses for each maturity group were similar, only the results of the analysis combining both maturity groups will be presented.

Table 2 displays EFs with a VIP greater than 1.9. Among the nine EFs selected for this criterion, all concerned the First Pod to Physiological Maturity phenophase: two from the First Pod to First Seed (FPFS), three from First Seed to End of Pod (FSEP) and four from End of Pod to Physiological Maturity (EPPM). The water stress index, cumulative evapotranspiration and duration had a VIP greater than 1.9 for both phenophases (FSEP and EPPM), with higher VIPs at EPPM. Regarding the cumulative daily solar radiation, only the EPPM phenophase was concerned.

Among selected EFs, the cumulative evapotranspiration during EPPM showed the highest VIP and had a strong positive impact on the observed yields (VIP = 2.86; β-coefficient = 0.14; $R^2$ = 0.64) (Figure 2a). The water stress index during EPPM had a strong negative impact on the observed yields (VIP = 2.82; β-coefficient = −0.15; $R^2$ = 0.53) (Figure 2b). The duration and cumulative daily solar radiation during EPPM presented a high VIP (VIP = 2.54 and VIP = 2.25, respectively) and impacted the observed yields positively (β-coefficient = 0.13 and β-coefficient = 0.1, respectively). No EF from phenophases prior to the First Pod stage showed a VIP greater than 1.9.

Among selected EFs, strong correlations (r > |0.8|) were observed. The cumulative evapotranspiration at EPPM was positively correlated with the cumulative solar radiation at EPPM (r = 0.89; $p < 0.001$) and negatively with the water stress index at EPPM (r = −0.84; $p < 0.001$). The water stress index and the phenophase duration were negatively correlated at both EPPM and FSEP (r = −0.94 and r = −0.85, respectively; $p < 0.001$).

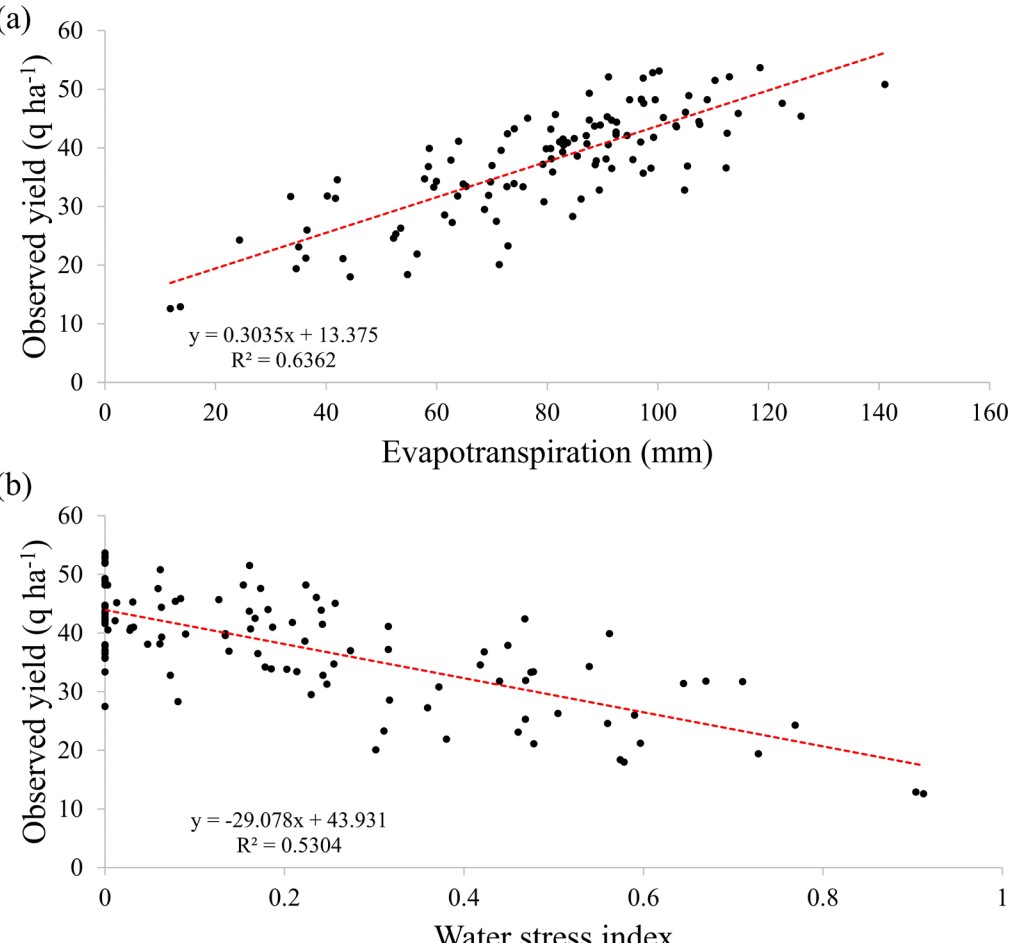

**Figure 2.** Relationship between observed soybean grain yield and the two first contributing eco-climatic factors from End of Pod to Physiological Maturity. (**a**) Sum of evapotranspiration. (**b**) Water stress index (0 = no stress, 1 = high stress). Water stress index and sum of evapotranspiration were simulated daily by the CROPGRO-soybean model.

*3.2. Main Eco-Climatic Factors Affecting Genotype by Environment Interactions*

The upper part of Figure 3 displays the frequency of each phenophase involved in the selected EFs for each meta-analysis. The results revealed that each phenophase, from Sowing to Physiological Maturity, contributed to GEI. The First Flower to First Pod (FFFP), First Pod to First Seed (FPFS) and First Seed to End of Pod (FSEP) were the most frequently detected phenophases. In PY-MG000 analyses, phenophases occurring after the First Pod stage were prevalent (100%).

Among the 48 selected EFs, solar radiation, maximum temperature and minimum temperature variables were the ones explaining GEI the most (Figure 3). In the SY-MG00 analysis, solar radiation variables after the First Pod stage (photothermal quotient, average and cumulative daily solar radiation) and minimum temperature variables (average minimum temperature, number of days below 10 °C and 15 °C) from Sowing to First Flower were selected. For SY-MG000, the selected variables were average temperature from Emergence to First Flower, then minimum temperature during FPFS and maximum temperature from First Seed to Physiological Maturity. For PY-MG00, minimum, average and maximum temperature were selected along the growth cycle. In the PY-MG000 analysis, mainly maximum and average temperature variables from First Seed to Physiological Maturity were selected.

| | | SEM | EMFI | FIFF | FFFP | FPFS | FSEP | EPPM |
|---|---|---|---|---|---|---|---|---|
| **Frequency of the phenophases involved in the selected eco-climatic factors** | Single year MG00 ○ | 17% | 8% | 8% | 8% | 25% | 17% | 17% |
| | Single year MG000 □ | 0% | 17% | 17% | 0% | 42% | 8% | 17% |
| | Pair of years MG00 ● | 8% | 8% | 8% | 42% | 8% | 8% | 17% |
| | Pair of years MG000 ■ | 0% | 0% | 0% | 0% | 8% | 67% | 25% |
| **Duration** | | | | | ○ | | | |
| **Water balance** | Cumulative potential evapotranspiration | | | | | □ | ○ | |
| | Cumulative precipitation | | □ | | | | ■ | |
| **Solar radiation** | Photothermal quotient | ○ | | | ● | ○□ | | ○ |
| | Average daily solar radiation | | | | | ○■ | ○■ | ○ |
| | Cumulative daily solar radiation | | | | | ○ | | |
| **Minimum temperature** | Number of days below 10°C | | ○ | | ● | | | |
| | Number of days below 15°C | | | ○ | | □● | | |
| | Average minimum temperature | ○● | | | ● | □ | | |
| **Temperature** | Average mean temperature | | | | □ | □ | ■ | |
| | Thermal amplitude | | □● | □ | | | ●■ | ●■ |
| **Maximum temperature** | Number of days above 30°C | | | | ● | | ■ | □● |
| | Number of days above 34°C | | | ● | | | ■ | |
| | Average maximal temperature | | | | | | □■ | □■ |
| **Nitrogen Stress** | | | | | | | ● | |
| **Photoperiod** | | | | | | | ■ | ■ |

**Figure 3.** Frequency of phenophases and repartition of the eco-climatic factors selected for their main impact on genotype by environment interactions in the different meta-analyses. Twelve eco-climatic factors were selected for the analyses on ○ single year by MG00, □ single year by MG000, ● pair of years by MG00 and ■ pair of years by MG000. (SEM: Sowing to Emergence; EMFI: Emergence to Flower Induction; FIFF: Flower Induction to First Flower; FFFP: First Flower to First Pod; FPFS: First Pod to First Seed; FSEP: First Seed to End Pod; EPPM: End Pod to Physiological Maturity). The double horizontal bars separate the growth phases: emergence, vegetative growth, reproductive growth and grain filling.

### 3.3. Effect of Eco-Climatic Factors Frequently Detected in GEI Analyses on Genotypic Responses

In this section, examples of genotype responses to contrasted levels of key EFs, based on their opposite β-coefficients, are displayed. Cv. 'ATACAMA' and 'SOLENA' showed a differential reaction to the average daily solar radiation during the FPFS phenophase (Figure 4a). When the cv. 'ATACAMA' average relative yield increased from 100% to 110% in environments with higher solar radiation (environments below 23 MJ m$^{-2}$ d$^{-1}$ versus environments above 23 MJ m$^{-2}$ d$^{-1}$), the cv. 'SOLENA' relative yield decreased from 102% to 93%. Moreover, cv. 'RGT SHOUNA' and cv. 'SULTANA' had contrasted reaction norms to the number of days with a minimum temperature below 15 °C (< 6.3 or ≥ 6.3 days) at the FPFS phenophase (Figure 4b). When the cv. 'RGT SHOUNA' relative yield increased from

98.8% to 94.5% with the number of days below 15 °C, the cv. 'SULTANA' relative yield decreased from 94.2% to 98.7%. Likewise, cv. 'RGT SIROCA' and cv. 'TIMOR PZO' were contrasted in their response to increased average minimum temperature during the FFFP phenophase (from 93% to 100% and from 104% to 99%, respectively) (Figure 4c). Finally, cv. 'OBELIX' and cv. 'SIRELIA' showed a differential reaction to the average maximum temperature during the FSEP phenophase (from 100% to 97% and from 96% to 104%, respectively) (Figure 4d).

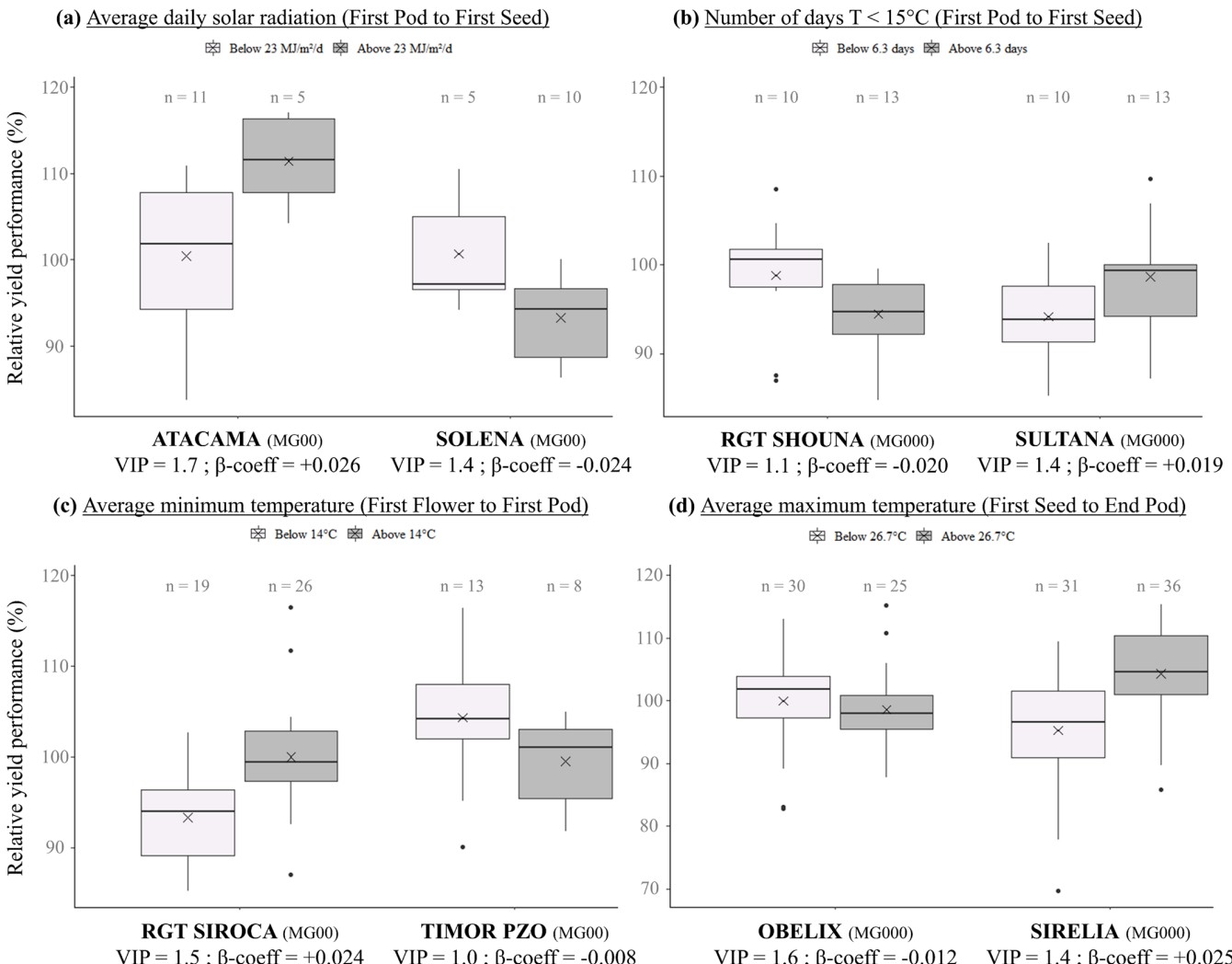

**Figure 4.** Relative yield performance of cultivars with contrasted response to different levels of selected eco-climatic factors frequently involved in GEI. (**a**) Relative yield of cv. 'ATACAMA' and cv. 'SOLENA' by average daily solar radiation during First Pod to First Seed (FPFS) phenophase. (**b**) Relative yield of cv. 'RGT SHOUNA' and cv. 'SULTANA' by number of days with a minimum temperature below 15 °C during First Pod to First Seed (FPFS) phenophase. (**c**) Relative yield of cv. 'RGT SIROCA' and cv. 'TIMOR PZO' by average minimum temperature during First Flower to First Pod (FFFP) phenophase. (**d**) Relative yield of cv. 'OBELIX' and cv. 'SIRELIA' by average maximum temperature during First Seed to End Pod phenophase. Cultivar relative yields were calculated as the ratio of cultivar yield on the performance checks average yield present in all trial/year combinations. Only analyses showing a VIP greater than 1 for the selected eco-climatic factors were retained. Average VIP and β-coefficient are given for each cultivar. The numeric value "*n*" above each box corresponds to the number of environments included in each box plot. Each box displays the median, upper and lower quartiles of the respective distribution. Box whiskers represent the maximum and minimum range excluding any extreme outliers (shown as dots).

**Table 2.** Eco-climatic factors with a VIP > 1.9 sorted by descending VIP attributed after PLS analysis. PLS analysis was performed on 112 trials conducted from 2017 to 2021, associated with 126 eco-climatic factors (i.e., environmental factors calculated between two developmental stages). The statistical model estimated that the five-components option was the optimal solution based on minimising RMSE. β-regression coefficients indicate whether the considered factor had a positive or negative effect on observed yields.

| EF | Description | VIP | β-Coeff | Std. Err. | *t*-Value | *p*-Value |
|---|---|---|---|---|---|---|
| ETsumEPPM | Sum of evapotranspiration from End Pod to Physiological Maturity (mm) | 2.86 | 0.14 | 0.02 | 7.26 | 0 |
| WSMNEPPM | Average water stress index from End Pod to Physiological Maturity | 2.82 | −0.15 | 0.02 | −7.08 | 0 |
| DurationEPPM | Duration of the period from End Pod to Physiological Maturity (days) | 2.54 | 0.13 | 0.02 | 7.30 | 0 |
| SRADsumEPPM | Sum of solar radiation from End Pod to Physiological Maturity (MJ/m$^2$) | 2.25 | 0.10 | 0.02 | 6.18 | 0 |
| WSMNFSEP | Average water stress index from First Seed to End Pod | 2.16 | −0.08 | 0.03 | −3.08 | 0 |
| ETsumFSEP | Sum of evapotranspiration from First Seed to End Pod (mm) | 2.08 | 0.05 | 0.02 | 2.91 | 0 |
| WSMNFPFS | Average water stress index from First Pod to First Seed | 2.02 | −0.08 | 0.02 | −4.00 | 0 |
| NSMNFPFS | Average nitrogen stress index from First Pod to First Seed | 1.91 | −0.10 | 0.04 | −2.59 | 0.01 |
| DurationFSEP | Duration of the period from First Seed to End Pod (days) | 1.91 | 0.05 | 0.02 | 2.46 | 0.02 |

## 4. Discussion

### 4.1. Eco-Climatic Factors Influencing Soybean Yields

The multi-year and empirical approach developed in this study allowed the ranking of 126 eco-climatic factors according to their relative impact on MG00 and MG000 soybean yields in a high-latitude context. These factors resulted from a combination of seven phenophases (the period between two developmental stages) and 18 environmental variables. The First Seed to Physiological Maturity (FSPM), encompassing the First Seed to End Pod (FSEP) and the End Pod to Physiological Maturity (EPPM) phenophases, appeared to be the most yield-influencing phenophases in our context. The FSPM phenophase, corresponding to the seed filling, was already identified in a long-term simulation study [46], but our results specified the prevalence of the EPPM phenophase. Herein, periods prior to the First Seed appeared to be less critical in determining soybean yields, as no eco-climatic factor from the vegetative period was prevalent in the analysis. Our results confirmed the predominance of the reproductive period in soybean yield establishment, where key yield components such as the number of seeds per pod and seed size are determined [62].

Regarding the First Seed to Physiological Maturity, the results revealed four main climatic factors: water stress index, evapotranspiration, period duration and cumulative daily solar radiation. The average values for the water stress index and potential evapotranspiration were 0.22 (from 0 to 0.91) and 80 mm (from 12 to 141 mm), respectively. Increasing water stress index is not favourable to yield (β = −0.08 and β = −0.15, respectively for FSEP and EPPM), contrary to evapotranspiration (β = 0.05 and β = 0.14, respectively for FSEP and EPPM). On the one hand, a water deficit is known to negatively impact physiological processes such as $CO_2$ assimilation, leaf senescence, xylem and phloem sap transportation and turgor maintenance [63] as well as $N_2$ fixation [35]. On the other hand, the evapotranspiration has been demonstrated to be well correlated to yield potential in many crops [64]. Overall, these two factors seem to impact in opposite ways the carbon and nitrogen metabolisms involved in the final seed weight, which is one of the key yield components [42]. To prevail yield loss and reach better yield potential, the use of reasoned irrigation during the seed filling period or soil tillage practices involved in soil

evapotranspiration mitigation (e.g., no or strip tillage) could be a lever to prevent water stress [45,46].

Both cumulative daily solar radiation and period duration at the grain-filling period had a positive influence on yield (positive β coefficient ranging from 0.1 to 0.13). The average values are 427 MJ m$^{-2}$ (from 290 to 605 MJ m$^{-2}$) and 21.1 days (from 17 to 25 days), respectively. Solar radiations are known to be the source of energy needed for carbon fixation, leading to plant biomass production [65]. The increase in solar radiation accumulation by the crop enhances global photosynthetic activity and, necessarily, the dry matter accumulation in the seeds, as demonstrated in maize by Daynard et al. (1971) [66]. The increase in the grain-filling duration allows crops to accumulate more solar radiation (r = 0.5; $p < 0.001$). Moreover, the EPPM duration is found to be well correlated to water stress (r = −0.94; $p < 0.001$), meaning that the intensity of drought stress is the main factor influencing grain-filling duration and thus solar radiation accumulation.

### 4.2. Major Eco-Climatic Factors Impacting Genotype by Environment Interactions

This study is the first attempt to identify the most prevalent Eco-climatic Factors (EFs) (i.e., combination of critical phenophases and climatic variables) driving Genotype by Environment Interactions (GEI) in MG00 and MG000 soybeans (Figure 3). An early soybean multi-environment trials database (112 environment over five years testing 57 cultivars) was built to unravel the GEI variance resulting from the contrasted contribution of a large number of EFs [20]. Transforming cultivar yield data into an interaction matrix revealed GEI sources and complexity, ensuring that they are not confounded by or linked with yield levels [22]. This differs from studies focusing on explaining yield performances per se or characterising environments to test their contribution to GEI effects afterwards [11,67,68]. The multi-year database on the one side (i.e., analyses on single years and pairs of years) and the contrasted networks for MG00 and MG000 on the other side were key to assess the diversity of GEI origins [19]. For instance, working on pairs of years allowed to test the genotype-by-year interaction effects, while single year analyses detected factors influencing genotype-by-location interaction effects. Other assets of our approach were the use of commercial cultivars and germplasm diversity tested each year and across years, revealing factors that are not addressed within the breeding process.

Single year analyses identified First Pod to First Seed as the prevalent phenophase for both maturity group (25% and 42%, respectively, for MG00 and MG000). Environmental stresses during pod setting are known to reduce the number of pods per plant, a major yield component [42,69,70]. The climatic variables affecting the GEI differed by maturity group. Our results are in light with previous studies [31,32,34] and revealed that, in MG000 soybeans, both the number of days with a minimum temperature below 15 °C (mean = 7.4, min = 0 and max = 14 days) and the average minimum temperature at FPFS (mean = 14.3, min = 10.9 and max = 20.1 °C) are crucial to explain GEI. For this EF, a genotypic contrast can be observed (Figure 4b). To further understand this contrasted germplasm behaviour in relation to low temperature, a growth chamber experiment should be designed to characterise the above-ground and below-ground plastic response to cold stress. Concerning MG00, the analysis revealed the importance of solar radiation variables (average and cumulative solar radiation and photothermal quotient) at FPFS for GEI (mean = 21.9, min = 14, max = 29 MJ m$^{-2}$ d$^{-1}$; mean = 285.5, min = 196.1, max = 371.3 MJ m$^{-2}$ d$^{-1}$; mean = 1.08, min = 0.73, max = 1.45 MJ m$^{-2}$ d$^{-1}$, respectively). This period corresponds to the establishment of the potential number of flowers, pods and seeds; they are highly demanding in energy (light intensity), especially in semi-determinate and indeterminate plants, where vegetative and reproductive growth occur concomitantly [42]. Solar radiation intensity around First Pod to First Seed is known to correlate with pod number and grain yield [8,71]. In a recent experiment, Müller et al. (2017) [72] demonstrated a genotypic variation of solar radiation interception capacity at vegetative stages in late soybeans. Herein, in early soybeans, we observed a genotypic variation of the response to different levels of solar

radiation, suggesting different capacities of cultivars to use limited available solar radiation efficiently (Figure 4a).

Analyses on pairs of years revealed contrasts in EF between MG00 and MG000 soybeans (Figure 3). These genotype-by-year interaction effects seemed to be more dependent on the early reproductive period in MG00 soybeans (mostly FFFP = 42%) when the late reproductive period is showing more GEI in MG000 soybeans (mostly FSEP = 67%). Interestingly, both cold stress (number of days with a minimum temperature below 10 °C and average minimum temperature) and heat stress (number of days above 30 °C) affect GEI for MG00 while not being correlated (r = 0.11). Selecting cultivars that are more tolerant to these stresses would limit flower abortion and would better control GEI in MG00 [73]. Concerning MG000, heat stresses at the FSEP phenophase were the most prevalent factors affecting GEI. During the reproductive period, the optimal temperature for seed yield varies between 22 and 24 °C [74]. Increasing the temperature from 30 °C to 35 °C during the First Seed to Physiological Maturity phenophase reduces the number of seeds per pod, the single seed weight, the seed weight per plant, the seed filling rate and the photosynthetic rate (decrease of 16%, 15%, 18%, 20%, 8% respectively) [75]. Regarding breeding for early soybeans, responses to heat stress should be considered as a relevant parameter. The genotypic response to daily maximum temperature that exists in the germplasm could be useful to select more adapted cultivars (Figure 4d). From this perspective, we would suggest building an environmental classification system to better control the complexity of GEI effects, as was already achieved in maize [76].

## 5. Conclusions

This study, combining multi-environmental trials and crop simulations, is the first attempt to identify and understand the Eco-climatic Factors (EFs) influencing yield levels on the one side and Genotype by Environment Interactions (GEI) for yield on the other side in MG000 and MG00 European soybeans. Yield levels for both maturity groups were mostly influenced by climatic factors from the grain-filling periods, with a minor impact of the vegetative and early reproductive stages. The most critical ones were water stress, evapotranspiration and quantity of solar radiation. Interestingly, EFs explaining GEI differed remarkably from those explaining yields, both in terms of critical phenophases and climatic factors. Moreover, the main EFs explaining GEI were contrasted between MG00 and MG000 soybeans. For MG00, cold stress during the vegetative growth and early reproductive period as well as solar radiation intensity during the seed setting were the main GEI-drivers. For MG000, cold stress during the reproductive period and high temperatures during the late seed filling period were the main GEI-drivers. Contrasted responses of cultivars to the main GEI-drivers were observed, revealing a genetic variability that could be directly exploited. In breeding, the promise is to deliver germplasm that outperforms the existing genetic material. Understanding the underlying factors driving the adaptation to the target environment offers elements to adapt breeding strategies. Breeders should screen their germplasm against the relevant GEI-drivers to improve yield stability. Knowing strengths and weaknesses of the available germplasm for these critical EFs will help breeders to make better choices of parental lines. This study has demonstrated the complexity of GEI and the diversity of contributing factors. Considering this complexity, it is now possible, by selecting the most frequent EFs, to define discriminating climatic scenarios (i.e., environmental classes) for GEI in northern European regions.

**Supplementary Materials:** The following supporting information can be downloaded at: https://www.mdpi.com/article/10.3390/agronomy13020322/s1, Table S1: Distribution of very early (MG000) and early (MG00) soybean cultivars grown over the four years of trials (2017–2020). Number of entries per trial and number of trials per year are summarized at the bottom. Table S2: Maturity groups 00 and 000 (MG00 and MG000, respectively) genetic coefficients used for the simulation of MG00 and MG000 environments, respectively.

**Author Contributions:** C.E., G.B., M.-P.F. and B.L. contributed to the conception and conduction of in silico experiments. L.L. produced the R script used for PLS regression. All authors have read and agreed to the published version of the manuscript.

**Funding:** This research received no external funding.

**Data Availability Statement:** Not applicable.

**Acknowledgments:** The "SOJA Terres Inovia-GEVES-Partenaires" are acknowledged for the open access trial dataset. The authors are grateful to Patrice Jeanson (Lidea Seeds) for his insights on the manuscript. We are thankful to Romain Armand and Chloé Girka (Institut Polytechnique UniLaSalle, Beauvais, France) for their help in mapping climatic conditions and trials.

**Conflicts of Interest:** The authors declare no conflict of interest.

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
