# Peer review of "Identification of Eco-Climatic Factors Driving Yields and Genotype by Environment Interactions for Yield in Early Maturity Soybean Using Crop Simulation"

_agronomy, doi:10.3390/agronomy13020322_

Round 1

Reviewer 1 Report

        Early maturity variety is one of the preconditions for expanding soybean to regions with higher latitudes and altitudes. In this manuscript, the authors identify the most prevalent Eco-climatic Factors (EFs) including combination of critical phenophases and climatic variables) driving Genotype by Environment Interactions (GEI) in MG00 and MG000 soybeans. They found that the yield levels for both early maturity groups were mostly influenced by climatic factors from the grain-filling periods, with a minor impact of the vegetative and early reproductive stages. The confirmation of the complexity of GEI and the diversity of contributing factors facilitates the understanding the underlying factors driving the adaptation of soybean to the target environments in the short-season regions.

      The manuscript can be improved by revising or confirming the following aspects:

1.   Brazil is now the No. 1 soybean producer and soybean is widely distributed in its tropical region. In fact, the successful breeding of Tropical Soybean in the major reason for Brazilian to develop their soybean industry. Therefore, the description in Lines 68-70 should be revised.

2.   In Line 79-81, the authors cited that low temperatures under 15°C are considered as a chilling stress in soybean. That may be true for pod setting and filling but not for germination and early seedling growth.

3.   The authors used the standards established by Fehr and Carviness (1971) to describe the developmental stages of soybean. However, they did not used the commonly-accepted descriptions like VE (emergence), R1 (beginning bloom), R7 (physiological maturity), etc. In addition, two new stages including Flower Induction and End of Pod are introduced. As I understand, flower induction is a very complicated process. It needs anatomical observation or even molecular investigation to confirm flowering induction process is completed or not. The End of Pod is also very difficult to determined. I suggest the authors to follow the terms and descriptions proposed by Fehr and Carviness. By the way, the reference 62 for the description was not correctly cited (Report number is missed).

4.   In a previous study (Song W et al. Analysing the effects of climate factors on soybean protein, oil contents and compositions by extensive and high-density sampling in China. Journal of Agricultural and Food Chemistry, 2016, 64(20): 4121-4130), it was found that diurnal temperature range (DTR) was the main factor that directly affected soybean protein and oil contents. Considering that protein and oil are the major compositions in soybeans, it is interesting to analyze if the DTR is the Eco-climatic Factors (EFs) for the yield for the early maturity soybeans in France. The reviewer notices that the authors used the thermal amplitude as a variable, but it was not clear it was daily or periodical data in the manuscript.

5.   There are several typing or editing mistakes in Line 245, 263 and 265.

6.   In Line 356, ‘pre-eminencecan be replaced by another word for easily understanding.

7.   In Line 442, ‘Figure 5d’ seems to be Figure 4d. Please confirm.

8.   There are many abbreviations of the key terms in the text. It will be more convenient for the readers if a collection of explanations of abbreviation is provided given that the journal allows to do that.

Reviewer 2 Report

Study entitled “Identification of eco-climatic factors driving yields and genotype by environment interactions for yield in early maturity soybean using crop simulation” investigated the role of ecoclimatic factors impacting on soybean yield. Overall, all the sections of MS are well written. However, authors need to improve the manuscript by making the indicated corrections below.

 Line 33: better to use symbol × instead of letter x. ?

Lines 95-96: remove the repetition of acronyms/abbreviations as they are explained in the first mentioned place.

Lines 144-145: add a reference

Lines 166-169: what is meant by “these generic cultivars were used to simulate phenophases in the different MG00 and MG000 environments” ? total of 57 cultivars from two groups were assessed ….which generic cultivar was used for which specific group ? or different cultivars were used? Could you please provide the table (can be in supplementary material) of cultivar coefficients used for genotypes in both groups.

Lines 245, 263, 265, : please check the reference. And check the references throughout MS

Line 277: Figure 2: Which method was used in DSSAT for simulating evapotranspiration? Considering water stress index … which stress index is presented here and correlated with yield

Figure 4: is it possible to increase the quality of figure and increase the size of text and numbers in it …please check

Lines 341-342: please add the description of box plot boundaries and clearly indicate the presented data in boxes and whiskers

Recheck the format of references and journal names, italicize scientific names i.e, Line 557, 588, 653 etc ..as well

Reviewer 3 Report

The presented article is devoted to the problem of crops spreading to the new territories and their adaptation to new environmental constraints, particularly, expansion of soybeans to North of Europe. This process requires the revealing of main eco-climatic factors influencing productivity, earliness and other agronomic characters. Though nowadays many experiments for revealing the influence of eco-climatic factors on plants agronomic characters expression are held (which is shown in the article), each crop has its own peculiarities of the interactions between genotypes and weather conditions. So, current investigation is obviously actual. Another difficulty of success breeding is the interaction between plants’ genotypes and complex of environmental characters. To analyze this interaction in frames of the experiment a huge dataset of plants characters and weather parameters was created. It is a great pity, that real pheno- fazes were not documented and authors had to simulate them using previously developed model. On the bases of this data, it was discovered that yield levels of the varieties belonging to two maturity groups were influenced mainly by weather conditions occurring during seeds formation. The most important factors appeared to be water stress, evapotranspiration and quantity of solar radiation. In spite of this disadvantage, obtained results can be used in practice of soybean breeding and help to select appropriate initial material for creation of new varieties.

At the same time I have some remarks.

1. Figures are too small. It is very difficult to digest them.

2. In the text many abbreviations are used. To my mind it would be useful to insert the table with the list of abbreviations, or exclude abbreviations (as I understand, the volume of the article is not limited).

3. It would be useful to describe the used method of pheno- fazes simulation in more details.
